# Time-periodic corner states from Floquet higher-order topology

Weiwei Zhu [1,4], Haoran Xue [2,4], Jiangbin Gong [1✉], Yidong Chong [2,3✉] & Baile Zhang [2,3✉]

The recent discoveries of higher-order topological insulators (HOTIs) have shifted the paradigm of topological materials, previously limited to topological states at boundaries of materials, to include topological states at boundaries of boundaries, such as corners. So far, all HOTI realisations have been based on static systems described by time-invariant Hamiltonians, without considering the time-variant situation. There is growing interest in Floquet systems, in which time-periodic driving can induce unconventional phenomena such as Floquet topological phases and time crystals. Recent theories have attempted to combine Floquet engineering and HOTIs, but there has been no experimental realisation so far. Here we report on the experimental demonstration of a two-dimensional (2D) Floquet HOTI in a three-dimensional (3D) acoustic lattice, with modulation along a spatial axis serving as an effective time-dependent drive. Acoustic measurements reveal Floquet corner states with double the period of the underlying drive; these oscillations are robust, like time crystal modes, except that the robustness arises from topological protection. This shows that space-time dynamics can induce anomalous higher-order topological phases unique to Floquet systems.

[1] Department of Physics, National University of Singapore, Singapore 117542, Singapore. [2] Division of Physics and Applied Physics, School of Physical and Mathematical Sciences, Nanyang Technological University, Singapore 637371, Singapore. [3] Centre for Disruptive Photonic Technologies, Nanyang Technological University, Singapore 637371, Singapore. [4] These authors contributed equally: Weiwei Zhu, Haoran Xue. ✉email: phygj@nus.edu.sg; yidong@ntu.edu.sg; blzhang@ntu.edu.sg

HOTIs are a class of recently discovered topological phases of matter that extend the standard framework of band topology[1–10]. For example, a 2D second-order topological insulator—unlike a 2D first-order topological insulator that supports one-dimensional (1D) topological edge states—hosts zero-dimensional corner states determined by nontrivial higher-order bulk topology (Fig. 1a). This generalised bulk-boundary correspondence predicts the existence of topological states at lower-dimensional boundaries (e.g., corners), allowing for the topological characterisation of many materials that would previously have been considered trivial, such as twisted bilayer graphene[11]. HOTIs have attracted great interest among fields ranging from condensed matter to photonics and acoustics. In particular, they have been realised in various classical 'metamaterial' systems[3–6,8–10], aided by the ease with which metamaterial properties can be tuned. These previously realised HOTIs have nontrivial structure only in spatial dimensions, limiting the 2D higher-order topology to two major classes characterised by a nontrivial quadrupole moment and a nontrivial polarisation, respectively.

Time is another dimension that can be used to generate interesting bandstructure features. Floquet systems have time-periodic Hamiltonians satisfying $H(t + T) = H(t)$, where $T$ is a driving period. Such systems can exhibit unconventional topological phases such as Floquet topological insulators[12–14], which have properties that do not exist in their static counterparts. For example, chiral edge states can exist in an anomalous Floquet topological insulator whose bulk bands all have zero Chern number[13], violating the standard bulk-edge correspondence principle. There have been many proposals to combine Floquet engineering with higher-order topology[15–23], but none has been realised previously. Recently, a HOTI has been implemented in a coupled-resonator lattice[24], which can be described as a Floquet system using a scattering matrix framework[25,26]; however, the topological phase implemented in that work was a quadrupole insulator, which belongs to the class of static HOTIs.

It has been proposed that Floquet topological states can be utilised to construct period-doubled oscillations that effectively break discrete time-translation symmetry. These oscillations are reminiscent of time crystals, but have different origins[27]: whereas, a time crystal requires quantum many-body interactions to stabilise the breaking of discrete time-translation symmetry[28,29], the period-doubled Floquet corner states are topologically protected by the space-time symmetries of the lattice. The observation of this phenomenon, in the original proposal[27], requires the coexistence of two distinct Floquet topological phases in a 1D system. The underlying topology, being first order rather than higher order, is also anomalous since it cannot be described with conventional 1D topological invariants.

Here, we experimentally demonstrate an acoustic Floquet HOTI exhibiting topological corner states protected by space-time symmetries (Fig. 1b). The time-dependent drive is simulated by periodic modulation along a spatial axis in a static 3D lattice, which is a common method of realising Floquet dynamics[14,30]. Unlike static HOTIs, the corner states in the Floquet HOTI can oscillate in time, with oscillation period either equal to, or double of, the driving period. This extends the concept of anomalous Floquet band topology, previously limited to first order, to higher order. The quasienergy bands have zero quadrupole moment in addition to zero polarisation; in a static system, such features would point to a topologically trivial phase. This anomalous Floquet higher-order topology allows for the coexistence of two distinct Floquet topological states, thus satisfying the condition to observe period-doubled oscillation. We further demonstrate the coexistence of Floquet corner states and Floquet chiral edge states, which constitutes a situation of hybrid topological protection. We perform a series of experiments to probe the various unusual dynamical properties that arise from the system's Floquet higher-order topology.

## Results

**Model**. There have been many theoretical models proposed to realise Floquet HOTIs[15–23]. Here, we adopt a simple tight-binding model consisting of a 2D bipartite lattice[23] whose time-periodic driving protocol is illustrated in Fig. 2a. The model preserves particle-hole symmetry and inversion symmetry. The driving protocol consists of four steps with equal duration $T/4$. In each step, each site only couples to one of its four neighbouring sites (i.e., the instantaneous system is dimerised). A global dimerisation is introduced by letting the coupling strength in one of the four steps differ from other three: the coupling strength is $\gamma$ for steps 1, 2 and 4, while for step 3 the coupling strength is denoted by $\theta$. By varying $\gamma$ and $\theta$, we obtain a phase diagram containing various topological phases (see Supplementary Information). The system can exhibit two bandgaps, near quasienergies zero and $\pi$. We call these the 'zero bandgap' and '$\pi$ bandgap', and the corner states in these bandgaps 'zero modes' and '$\pi$ modes', respectively. All the quasienergy bandgaps that accommodate corner states are associated with zero quadrupole moment and zero polarisation, which is fundamentally different from previously studied static HOTIs (see Supplementary Information for a discussion of the topological characterisation).

**Floquet HOTI with $\pi$ corner modes**. We first consider the case $\theta = \gamma = 0.841\pi$. The quasienergy spectrum is plotted in Fig. 2b. There is a $\pi$ bandgap, which can host corner states, whereas the zero bandgap is closed. To realise this 2D Floquet model, we implement a 3D acoustic lattice (Fig. 2c) with one axis ($z$) playing the role of time[14,30]. In all the following calculations and

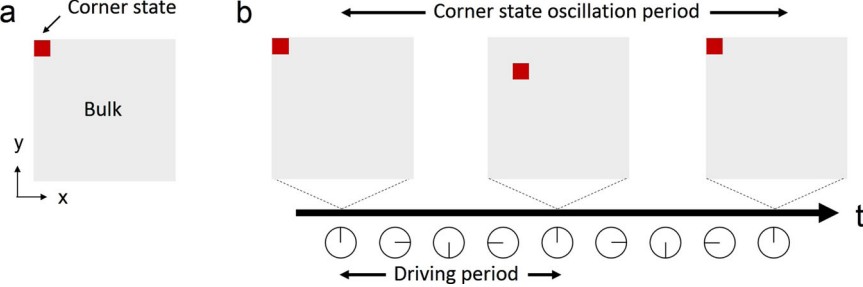

**Fig. 1 Schematic comparison between static and Floquet higher-order topological insulators. a** In a conventional second-order topological insulator, a corner state is localised at the corner with a time-invariant spatial distribution. **b** In a time-periodically driven, or Floquet, second-order topological insulator, the corner state oscillates in time near the corner. The corner state oscillation period can be different from the driving period. The schematic illustrates the scenario of a doubled period.

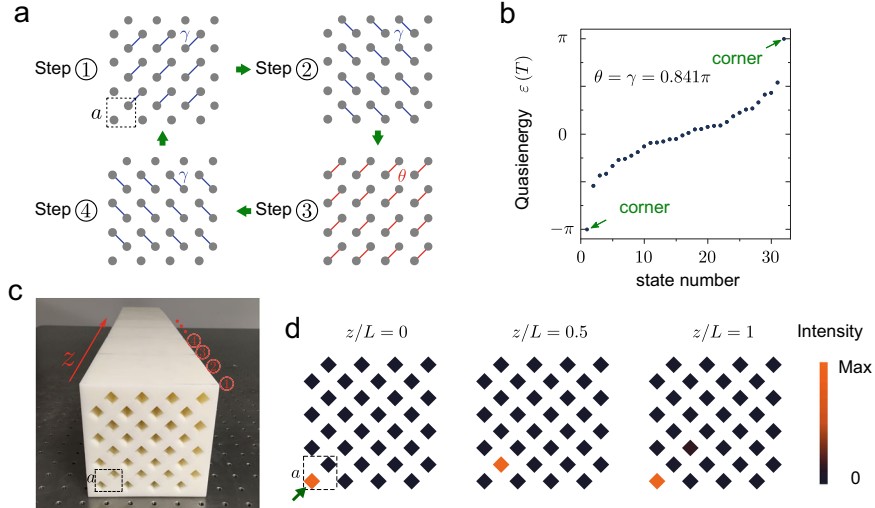

**Fig. 2 Design and construction of a Floquet higher-order topological insulator in an acoustic lattice. a** Tight-binding model and driving protocol. The drive consists of four steps with equal duration $T/4$, where $T$ is the driving period. The order of the four steps is indicated by the green arrows. In each step, one lattice site couples to one of its four neighbouring sites. The coupling strength is $\gamma$ in steps 1, 2 and 4, and $\theta$ in step 3. The dotted square in step 1 indicates the unit cell. **b** Numerically obtained quasienergy spectrum with $\theta = \gamma = 0.841\pi$. **c** Photo of the fabricated acoustic structure that realises the tight-binding model in **a**. Here the $z$ axis plays the role of time. The lattice constant $a = 20\sqrt{2}$ mm. The coupling strength $\theta = \gamma = 0.841\pi$ is accomplished by thin connecting channels that are not visible in the photo. **d** Measured acoustic intensity distributions at different evolution distances at 8000 Hz. The green arrow indicates the excitation position at $z/L = 0$.

demonstrations, we choose the lattice constant $a = 20\sqrt{2}$ mm, and four unit cells extend along both the $x$ and $y$ directions. Each site in the tight-binding model corresponds to a square air-hole waveguide with side length $l = 10$ mm surrounded by hard acoustic boundaries. The coupling between two adjacent sites is accomplished by placing, between two adjacent square wave-guides, a few thin connecting channels (these are not visible in Fig. 2c; see Supplementary Information for the design). By modulating the placement of the connecting channels along $z$, we realise an effective time-periodic driving following the protocol in Fig. 2a. In this construction, we take $L = 336$ mm as the modulation period along $z$. The sample in Fig. 2c has length $3.5L$. The coupling strength can be adjusted by altering the number of thin connecting channels; for example, by setting 12 connecting channels, $\theta = \gamma = 0.841\pi$ can be satisfied at 8000 Hz (see Supplementary Information for the numerical determination of the coupling strengths).

To demonstrate the dynamical properties of the $\pi$ corner modes, a speaker is placed at $z = 0$ on the lower-left corner (indicated by a green arrow in Fig. 2d). The acoustic pressure at different propagation distances is recorded by a microphone (see "Methods"). Figure 2d shows the measured evolution of the corner states, revealing strong localisation around the lower-left corner. The intensity oscillates between the two sublattices near the corner, each taking half a period, which is a characteristic feature of $\pi$ modes. These experimental observations are consistent with simulation results (see "Methods" and Supplementary Information), thus verifying the existence of the $\pi$ modes. Although we have only presented results for frequency 8000 Hz, these dynamical properties of the corner states are observed in a broad frequency range from 7500–8300 Hz (see Supplementary Information).

**Anomalous Floquet HOTI with 0 and $\pi$ corner modes simultaneously**. To further explore the properties of the lattice, we take different coupling strengths $\gamma$ and $\theta$. We consider the effects of

reducing the number of connecting channels in step 3 from 12 to 8, such that the coupling strength $\theta$ is reduced correspondingly to $0.568\pi$ while $\gamma = 0.841\pi$ is maintained at 8000 Hz. The resulting quasienergy spectrum is shown in Fig. 3a. In this case, both the $\pi$ bandgap and zero bandgap are open. The numerically obtained eigenmode profiles in Fig. 3b, c confirm the existence of zero modes and $\pi$ modes localised at the corners. The $\pi$ modes oscillate between two sublattices, consistent with Fig. 2d. Despite moderate changes over time, the zero modes mainly concentrate in one sublattice, similar to corner states in static HOTIs.

We then fabricated another experimental sample meeting the condition described in the previous paragraph ($\theta = 0.568\pi$ and $\gamma = 0.841\pi$). The resulting dynamics, shown in Fig. 3d, is very different from the previously studied case that had only $\pi$ modes present. Under corner excitation, the acoustic intensity is localised around the corner, but the mode profile does not repeat itself after one driving period, as is evident by comparing the acoustic intensities at $z/L = 0$ and $z/L = 1$. Instead, a doubled period is observed by comparing acoustic intensities at $z/L = 0$ and $z/L = 2$. The period doubling comes from the superposition of zero modes ($|0\rangle$) and $\pi$ modes ($|\pi\rangle$)—e.g., $a|0\rangle + b|\pi\rangle$, which evolves to another state after one driving period $U_L(a|0\rangle + b|\pi\rangle) = (a|0\rangle - b|\pi\rangle)$ and comes back to itself at two periods $U_{2L}(a|0\rangle + b|\pi\rangle) = (a|0\rangle + b|\pi\rangle)$, where $U_L$ is the evolution operator over one period. This period doubling feature has previously been predicted for the breaking of discrete time-translation symmetry with topological states[27], and is a striking outcome of the coexistence of zero and $\pi$ modes and the anomalous Floquet higher-order topology.

**Hybrid Floquet HOTI with 0 corner modes and $\pi$ chiral edge modes**. The zero and $\pi$ modes that we have observed are protected by the higher-order topology of the zero bandgap and $\pi$ bandgap, respectively. The topological properties of these two bandgaps can be separately controlled by tuning the coupling strengths $\gamma$ and $\theta$. Interestingly, it is possible for one bandgap to

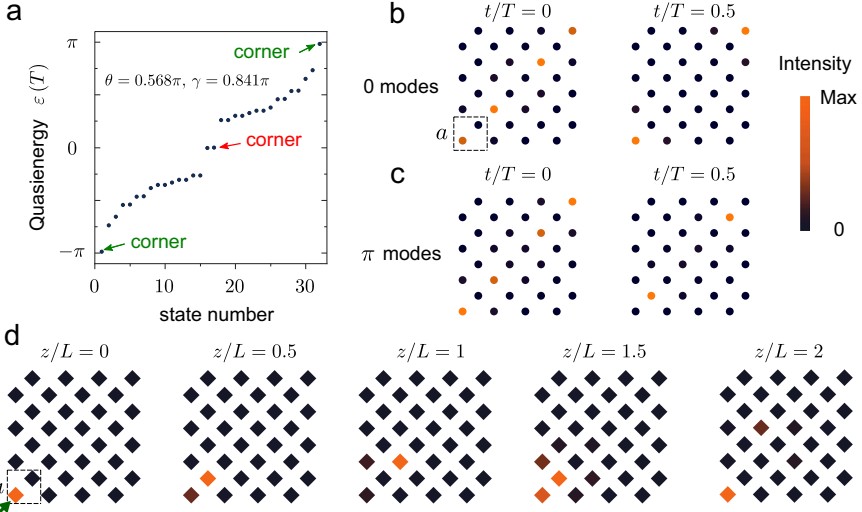

**Fig. 3 Demonstration of period doubling of Floquet corner states. a**, Numerically obtained quasienergy spectrum with $\theta = 0.568\pi$ and $\gamma = 0.841\pi$. **b** Numerical eigenmode profiles for the zero modes. **c** Numerical eigenmode profiles for the $\pi$ modes. **d** Measured acoustic intensity distributions at different evolution distances at 8000 Hz. The green arrow indicates the position of excitation at $z/L = 0$.

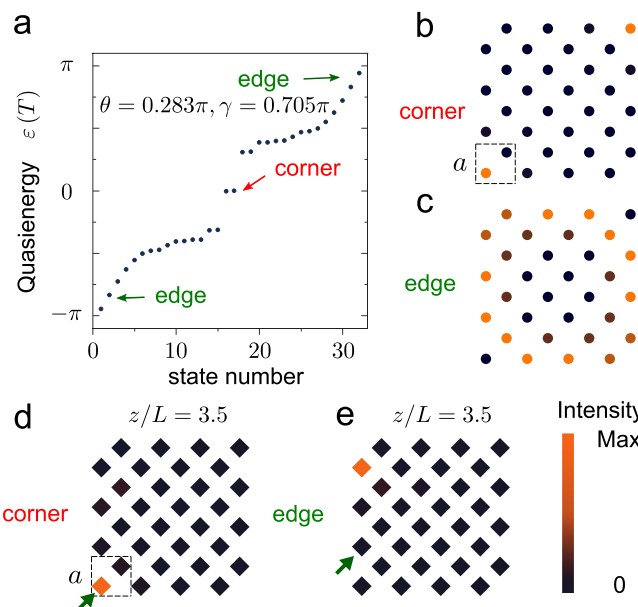

**Fig. 4 Demonstration of coexisting corner states and chiral edge states. a** Numerically obtained quasienergy spectrum with $\gamma = 0.705\pi$ and $\theta = 0.283\pi$. **b** Numerical eigenmode profiles for the corner states. **c** Numerical eigenmode profiles for the chiral edge states. **d** Measured acoustic intensity distribution at $z/L = 3.5$ at 8000 Hz with corner excitation. The green arrow indicates the position of excitation at $z/L = 0$. **e** Measured acoustic intensity distribution at $z/L = 3.5$ at 8000 Hz with edge excitation. The green arrow indicates the position of excitation at $z/L = 0$.

have nontrivial first-order topology, while the other exhibits higher-order topology. In that case, two types of protected boundary states—corner states and chiral edge states—can simultaneously exist. To study this phenomenon, we reduce the number of connecting channels in driving steps 1, 2 and 4 to 10 (so that $\gamma = 0.705\pi$) and that in driving step 3 (so that $\theta = 0.283\pi$). The numerically obtained quasienergy spectrum in Fig. 4a shows that the zero bandgap now hosts zero corner modes

(due to higher-order topology), while the $\pi$ bandgap is spanned by gapless chiral edge states (due to first-order topology). The calculated eigenmode profiles in Fig. 4b, c confirm that these are indeed coexisting corner states and chiral edge states, subject to their respective topological protection.

We fabricated a sample that meets these conditions ($\gamma = 0.705\pi$ and $\theta = 0.283\pi$). To probe the corner states and edge states separately, we conducted two measurements with different excitations, whose results are plotted in Fig. 4d, e. In the first measurement (Fig. 4d), the source is placed at the lower-left corner (indicated by the green arrow) at $z/L = 0$. In this case, the corner state is excited and the acoustic intensity is found to be localised at the corner after an evolution of 3.5 driving periods. In the second measurement (Fig. 4e), the excitation occurs along the left edge (indicated by the green arrow) at $z/L = 0$. The chiral edge state then propagates along the edge unidirectionally and moves up by around two lattice constants after an evolution of 3.5 driving periods. These observations provide direct evidence of the coexistence of Floquet corner states and Floquet chiral edge states, induced by Floquet engineering.

## Discussion

We have proposed and experimentally demonstrated a Floquet HOTI in an acoustic lattice. The Floquet higher-order topology is tied to inversion symmetry and particle-hole symmetry (see Supplementary Information), and gives rise to unusual dynamical properties not found in static HOTIs. The coexistence of zero modes and $\pi$ modes is reminiscent of an edge-state-based time crystal, except that the robustness in this case is tied to topological protection instead of many-body interactions[27]. These Floquet modes, if reproduced in a quantum lattice, may also find applications in measurement-based quantum computing[31]. The coexistence of Floquet corner states and Floquet chiral edge states may be useful for state transfer[32]. These results extend the concept of anomalous Floquet band topology from first order to higher order, substantially expanding the scope of higher-order topological phases (which, in 2D, had been limited to nontrivial quadrupole moment or polarisation). Although the concept has been demonstrated on an acoustic platform, similar models can also be realised in photonic systems such as coupled ring resonators[24,26] and laser-written optical waveguides[14] where the

effects of non-Hermiticity and nonlinearity can be more easily studied, or even in real time-dependent systems[33].

## Methods

**Numerical simulation**. The quasienergy spectra in the main text (i.e., Figs. 2b, 3a–c and 4a–c) are obtained from tight-binding calculations, using coupling parameters extracted from Comsol simulations. The procedure for retrieving the coupling parameters is detailed in Sec. VII of the Supplementary Information. The validity of these spectra is verified by full-wave simulations of 3D acoustic structures with periodic boundary conditions along $z$. At a fixed frequency (8000 Hz in the demonstration), the quasienergy ($k_z$) band can be obtained by restricting our attentions to the modes propagating in the $+z$ direction. The numerical field distributions given in Fig. S13 and Fig. S14 in the Supplementary Information are obtained from Comsol Multiphysics (pressure acoustic module). The boundaries of the 3D printing materials (photosensitive resin) are modelled as rigid acoustic walls due to the large impedance mismatch with air (density $\rho = 1.29$ kg/m$^3$ and sound speed $v = 343$ m/s). In all full-wave simulations, the models have the same size as the ones used in experiment ($z/L = 3.5$). The air boundaries at $z/L = 0$ and $z/L = 3.5$ are set to be radiation boundaries with an incident field applied to the lower-left corner or one site on the left edge at $z/L = 0$.

**Sample fabrication**. All samples are fabricated through a stereolithography apparatus with $\approx 0.1$ mm resolution. To measure the acoustic intensity at different positions along $z$, each sample is divided into six pieces with cutting positions $z/L \in \{0, 0.5, 1, 1.5, \ldots, 3.5\}$. These small pieces are fabricated separately and then assembled into the experimental sample.

**Experimental measurement**. All experiments are conducted using a similar scheme. An acoustic wave is generated by a loudspeaker and guided into one lattice site at $z/L = 0$ through a small tube. The output signals are recorded by a microphone (Brüel&Kjaer Type 4182) that sweeps all the sites at the output plane. The measured signals are processed by an analyser system (Brüel&Kjaer 3160-A-022 module) to obtain the frequency-resolved spectrum. In all figures showing experimental results, the data are normalised to the maximal value in the plot.

**Reporting summary**. Further information on research design is available in the Nature Research Reporting Summary linked to this article.

## Data availability

The experimental data are available in the data repository for Nanyang Technological University at https://doi.org/10.21979/N9/YBSECE. Other data that support the findings of this study are available from the corresponding authors on reasonable request.

## Code availability

All numerical codes are available from the corresponding authors on reasonable request.

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

## Acknowledgements

B.Z. and Y.C. acknowledge funding support by the Singapore Ministry of Education Academic Research Fund Tier-3 (Grant No. MOE2016-T3-1-006) and Tier 2 (Grant No. MOE2019-T2-2-085). J.G. acknowledges funding support by the Singapore Ministry of Education Academic Research Fund Tier-3 Grant No. MOE2017-T3-1-001 (WBS No. R-144-000-425-592) and by the Singapore NRF Grant No. NRF-NRFI2017-04 (WBS No. R-144-000-378- 281). We are grateful to H.-X. Sun, R. Bomantara and S. Mu for helpful discussions.

## Author contributions

W.Z. and H.X. contributed equally to this work. W.Z. and H.X. carried out the simulation. H.X. and W.Z. designed and performed the experiment. J.G., Y.C. and B.Z. supervised the project. All authors contributed extensively to the proposal of idea, the interpretation of the results and the writing of the paper.

## Competing interests

The authors declare no competing interests.
