## [Peer Review File · Nature Communications]

Time-periodic corner states from Floquet higher-order topologyEditorial Note: This manuscript has been previously reviewed at another journal that is not operating a transparent peer review scheme. This document only contains reviewer comments and rebuttal letters for versions considered at Nature Communications.

Reviewers' Comments:

Reviewer #1:

Remarks to the Author:

I'm still uncertain whether the 2d Floquet picture is the best way to analyze this specific system, but I agree that it might be not fair to challenge the novelty of this work from this perspective based on the new references offered. Since my main concern is addressed, there is no reason to refuse this manuscript in principle now. Below are some additional comments and some suggestions about revisions.

Honestly, I still feel that this system might be understood by analyzing the 3D structure directly. The corner states might be attributed to some hinge states in the 3D structure. I guess when the authors extract either bulk band structures or edge states, such as those presented in Fig.3 using COMSOL, they do have to treat it as a 3D structure by solving an eigenproblem with periodic boundaries and a given k_z , which is the quasi energy that should be solved in the Floquet picture.

I find that the subsection Numerical simulation in Method is not clear enough. I think obtaining the spectrum and mode profiles in Fig. 3 requires solving an eigenproblem. It seems that this is not the situation described in Method. I suggest the authors further clarify this point.

In the revised SI, the authors add Section VI and Figure R2 on the robustness of the boundary states against the disorders. I wonder whether Fig R2 is obtained via full-wave simulation or just solving a tight-binding model. I suggest the authors clarify their method here.

I guess it might be done by solving a tight-binding model and I understand that a full-wave simulation is really demanding. I agree that the original tight-bind model has a particle-hole symmetry that pins the quasienergy of the corners states. However, I don't think the full-wave system still has that symmetry. In my mind, nothing can pin that quasienergy in real acoustic systems, which is actually the momentum along the z -direction. I have no idea how to improve this part but I think since we agree that this system doesn't have filling anomalies, this part isn't that critical. This paragraph is just a discussion about some of my concerns and I don't require any revision on this part.

Reviewer #2:

Remarks to the Author:

In this work, Zhe et al report the first experimental demonstration of an anomalous Floquet higher order topology. This work considers a 3D acoustic lattice with modulation along the z axis (effectively the time axis). The experimental results agree qualitatively with the numerical simulations and model. This work extends the concept of anomalous Floquet band topology to higher dimension, i.e., a higher-order topological insulator with zero quadrupole moment and zero polarization. In all, this work is well-written, the results are well-organized and easy to follow. In my opinion, the novelty of this work is enough to guarantee a publication in Nature Communications.

I have only a few minor comments:

1. CROW systems are also treated as Floquet systems in some literatures. In Nature Photonics volume

- 13, 692 (2019), an experimental demonstration of HOTI is presented with CROW systems. I think the difference from the current work should be cleared.
2. This work uses a similar structure as Phys. Rev. Research 1, 033149 (2019) (or maybe other similar works), the contribution of previous works should be acknowledged. Since this work is now transferred to nature communications, the number of reference is not that limited.
 3. The coupling between different waveguides are introduced by thin channels, where the reflection is tuned to be small. However, there reflection is still around 0.1. Is it possible to further reduce the refecton with such a setup?
 4. When exciting the system, we actually fix the incident frequency not the quasi-energy, which means in principal all the eigenstates can be excited. While the experiments show good localization behavior. Any reason why? Where are the bulk states?
 5. Following Q4, the eigenstates in Figs. 4b and 4c have almost no overlap with each other, and thus one can selectively excite either state. I guess this (no overlap) is not a universal behavior. With some other parameters, the corner states may not be completely isolated from the edge states, then one cannot selectively excite each one.

Reviewer #3:

Remarks to the Author:

The paper "Time-periodic corner states from Floquet higher-order topology" by Zhu et al. provides an experimental demonstration of a Floquet HOTI exhibiting topological corner states protected by space-time symmetries. The experiment uses a 3D acoustic lattice with the z-axis playing the role of time. The authors claim that this is the first experimental realization of this theoretically proposed phenomenon, which makes the study valuable.

The main achievement of the paper is the experimental realization and observation of a HOTI. I, therefore, expect the advances reported on in the paper to be of interest mainly to experimentalists, but also to motivate theorists working on band topology. The experiment does not report anything exotic, or inconsistent with the underlying theory -- it merely provides an independent justification for HOTI in a specific physical system (3D acoustic lattice).

From a theoretical perspective, the paper does not provide a major new finding/discovery. However, a pioneering experimental realization itself can be valuable (and would justify publication in Nat Communications), provided it opens up a new avenue for further advances in the field.

Questions/Suggestions/Comments:

- the authors speak of a "hybrid topological protection induced by Floquet driving" -- I am a bit worried that the Floquet drive here is just a mathematical interpretation for the physics in the system. In reality, there is no time-periodic modulation applied to the system. Therefore, I urge the authors to do their best and avoid misleading formulations like the above, or at least make it clear that this is an interpretation.

- several times in the manuscript, the non-equilibrium character of the phenomenon is brought up. I would like to bring the authors' attention to the fact that not every time dependence results in nonequilibrium phenomena -- sometimes time is just an illusion of the reference frame. This is particularly easy to see in the present system where energy is conserved at all times [the system is a static 3D lattice with the z-axis playing the role of time]. I think all these statements should be

corrected.

- although I am not an expert in the field of experimental acoustics, I am surprised by the way the experimental data is presented in the main text. E.g., in Fig 3 panels b,c are theoretical data, while panel d is supposed to be the experimental data. There are basically no units for the measurement, the colorbar goes from zero to Max, and the Max value is not easy to find. This applies also to Fig 4. Naively I would expect experimental data to have noise/uncertainty, the effects of which can/should be discussed, etc.

- Similar to the issue above -- the data for the spectrum presented in the main text (Fig 3a and Dif 4a) comes from a numerical simulation. The experimental results would make much more compelling evidence for the realization of a HOTI, if the spectrum were measured in a realistic experiment. I acknowledge the answer to the corresponding question by Reviewer 2, yet I urge the authors to seriously consider developing techniques or measuring the bulk spectrum in acoustic systems in future studies.

- The study on the effect of disorder in the SI, Sec VI, can ideally be done in the experiment itself, through imperfections in the waveguide. So far Fig S8 is a theoretical simulation, and HOTIs have been analyzed by theorists before. It would be nice to test the robustness to disorder experimentally.

Response Letter to Reviewers

We are grateful for the constructive comments on this manuscript (NCOMMS-21-30287-T) from all three reviewers.

In the text below, reviewer comments are quoted in *Italics* and followed by our detailed response. We have also revised the manuscript and the Supplemental Material based on the reviewer comments, and these updates are highlighted in blue in those files. In the text below, the references to these updates are highlighted in blue and also by a vertical red line in the left margin.

GENERAL COMMENTS FROM 1st REVIEWER:

I'm still uncertain whether the 2d Floquet picture is the best way to analyze this specific system, but I agree that it might be not fair to challenge the novelty of this work from this perspective based on the new references offered. Since my main concern is addressed, there is no reason to refuse this manuscript in principle now. Below are some additional comments and some suggestions about revisions.

Response from Authors:

We thank Reviewer #1 for accepting the novelty of our work. Let us first clarify why the 2D Floquet picture is the appropriate way to analyze this specific system.

We use the sample with only π corner modes (Fig. 2 in the main text) as an example. Figure R1(a) shows the full eigenvalue spectrum of the 3D structure obtained from COMSOL simulations. As can be seen, there is no complete band gap. The band of hinge states (corresponding to the π modes in the 2D Floquet picture), plotted in red, have substantial overlap with the bulk states. This makes it challenging to analyze the topological characteristics of the bandstructure.

The Floquet picture clarifies things by only considering those modes propagating along one direction (e.g., +z), exploiting the fact that such waveguides exhibit negligible mixing between modes moving in +z and -z. Figure R1(b) shows the 3D bandstructure for only modes propagating in +z. At a fixed frequency, the system supports a quasi-energy (k_z) band gap. As explained in the main text, once we go to the 2D Floquet picture, we can understand the topological characterization of these gaps.

Figure R1 | Spectrum of the 3D structure with periodic boundary condition along z direction and open boundary condition along x and y direction. (a) All the eigenstates. (b) Those eigenstates propagating along +z direction. The corner states are colored in red.

SPECIFIC COMMENTS FROM 1st REVIEWER:

1st Reviewer -- Comment 1:

Honestly, I still feel that this system might be understood by analyzing the 3D structure directly. The corner states might be attributed to some hinge states in the 3D structure. I guess when the authors extract either bulk band structures or edge states, such as those presented in Fig.3 using COMSOL, they do have to treat it as a 3D structure by solving an eigenproblem with periodic boundaries and a given k_z , which is the quasi energy that should be solved in the Floquet picture.

Response from Authors:

The reviewer is certainly correct that we need to treat the system as a 3D structure when doing COMOSL simulations, and the corner states in the Floquet picture indeed manifest as hinge states in the 3D band structure (as shown in Figure R1, above). However, as we have argued, it is difficult to analyze these states directly from the point of view of static 3D band topology. Instead, they are best thought of as arising from the 2D Floquet picture, by considering eigenstates propagating along (say) $+z$, as shown in Figure R1(b).

1st Reviewer -- Comment 2:

I find that the subsection Numerical simulation in Method is not clear enough. I think obtaining the spectrum and mode profiles in Fig. 3 requires solving an eigen problem. It seems that this is not the situation described in Method. I suggest the authors further clarify this point.

Response from Authors:

We thank the reviewer for the suggestion. The spectra presented in the main text are obtained from the tight-binding model with coupling parameters extracted from Comsol simulations (see Sec. VII in the Supplementary Information for the procedure). We have verified that the tight-binding results are the same as those directly obtained from Comsol eigensolvers (like the one shown in Figure R1(b)). The numerical field distributions in Fig. S13 and Fig. S14 in the Supplementary Information are obtained from a Comsol frequency domain solver. In the new version, we have revised the relevant Methods passage as follows:

“The quasienergy spectra in the main text (i.e., Fig. 2b, Fig. 3a-c and Fig. 4a-c) are obtained from tight-binding calculations, using coupling parameters extracted from Comsol simulations. The procedure for retrieving the coupling parameters is detailed in Sec. VII of the Supplementary Information. The validity of these spectra is verified by full-wave simulations of 3D acoustic structures with periodic boundary conditions along z . At a fixed frequency (8000 Hz in the demonstration), the quasienergy (k_z) band can be obtained by restricting our attentions to the modes propagating in the $+z$ direction. The numerical field distributions given in Fig. S13 and Fig. S14 in the Supplementary Information are obtained from Comsol Multiphysics (pressure acoustic module).”

1st Reviewer -- Comment 3:

In the revised SI, the authors add Section VI and Figure R2 on the robustness of the boundary states against the disorders. I wonder whether Fig R2 is obtained via full-wave simulation or just solving a tight-binding model. I suggest the authors clarify their method here.

I guess it might be done by solving a tight-binding model and I understand that a full-wave simulation is really demanding. I agree that the original tight-bind model has a particle-hole symmetry that pins the quasienergy of the corners states. However, I don't think the full-wave system still has that symmetry. In my mind, nothing can pin that quasienergy in real acoustic systems, which is actually the momentum along the z -direction. I have no idea how to improve this part but I think since we agree that this system doesn't

have filling anomalies, this part isn't that critical. This paragraph is just a discussion about some of my concerns and I don't require any revision on this part.

Response from Authors:

Those results are obtained from a tight-binding calculation. We have added in the caption of Fig.S8 that “The results are obtained from a tight-binding calculation” to clarify this point.

We agree with the reviewer that the momentum along the z-direction cannot be pinned at 0 or $\pm \pi/L$. However, the system can support a more general particle-hole symmetry which meets $CH(k, z)C = k_z^0 - H^*(-k, z)$ and the momentum along z is pinned at k_z^0 or $k_z^0 \pm \pi/L$. The shift k_z^0 comes from the on-site potential of each acoustic waveguide, and it does change the eigenstates of the system, so we can observe the phenomena in experiment. Note that at a fixed frequency, the shift k_z^0 is almost a constant.

GENERAL COMMENTS FROM 2st REVIEWER:

In this work, Zhu et al report the first experimental demonstration of an anomalous Floquet higher order topology. This work considers a 3D acoustic lattice with modulation along the z axis (effectively the time axis). The experimental results agree qualitatively with the numerical simulations and model. This work extends the concept of anomalous Floquet band topology to higher dimension, i.e., a higher-order topological insulator with zero quadrupole moment and zero polarization. In all, this work is well-written, the results are well-organized and easy to follow. In my opinion, the novelty of this work is enough to guarantee a publication in Nature Communications.

Response from Authors:

We thank the reviewer for supporting the novelty of this work.

SPECIFIC COMMENTS FROM 2st REVIEWER:

2st Reviewer -- Comment 1:

1. CROW systems are also treated as Floquet systems in some literatures. In Nature Photonics volume 13, 692 (2019), an experimental demonstration of HOTI is presented with CROW systems. I think the difference from the current work should be cleared.

Response from Authors:

We thank the reviewer for the suggestion. The work in *Nature Photonics* 13, 692 (2019) realized a photonic quadrupole topological insulator in a CROW system. As the reviewer noted, it is possible to model CROW lattices as Floquet systems (by treating the scattering matrices as pseudo-time evolution operators). However, in that paper the CROW lattice was used only to implement a HOTI that can be realized in static systems (i.e., a quadrupole insulator). By contrast, the Floquet HOTI reported in our work has no static counterpart.

In the revised text, we cite *Nature Photonics* 13, 692 (2019) as Ref. [24], and discuss its significance in paragraph 2, page 1, as follows:

“Recently, a HOTI has been implemented in a coupled resonator lattice [24], which can be described as a Floquet system using a scattering matrix framework [25, 26]; however, the topological phase implemented in that work was a quadrupole insulator, which belongs to the class of static HOTIs.”

2st Reviewer -- Comment 2:

2. This work uses a similar structure as *Phys. Rev. Research* 1, 033149 (2019) (or maybe other similar works), the contribution of previous works should be acknowledged. Since this work is now transferred to nature communications, the number of reference is not that limited.

Response from Authors:

We thank Reviewer #2 for the suggestion. We have added *Phys. Rev. Research* 1, 033149 (2019) as ref. [30] to the reference list.

2st Reviewer -- Comment 3:

3. The coupling between different waveguides are introduced by thin channels, where the reflection is tuned to be small. However, there reflection is still around 0.1. Is it possible to further reduce the reflection with such a setup?

Response from Authors:

Yes, it is possible to further reduce the reflection by further reducing the width of coupling channels and increasing the number of coupling channels, such that the acoustic field in one waveguide can slowly couple to another waveguide while keeping the propagating direction. However, in acoustic systems, a thinner width of channels will lead to more loss. To satisfy this tradeoff, we considered a reflection of < 0.1 to be sufficiently small in the experiment design.

2st Reviewer -- Comment 4:

4. When exciting the system, we actually fix the incident frequency not the quasi-energy, which means in principal all the eigenstates can be excited. While the experiments show good localization behavior. Any reason why? Where are the bulk states?

Response from Authors:

The corner states and bulk states have different field profiles and are well-separated in space. When we excite at the corner site, the excitation has strong overlap with the corner states, which are thus efficiently excited. Let us use the third sample in the main text as an example. Figure R2 shows the profile of the sum of probabilities of all bulk states. The probability at two corners can be ignored, so the corner excitation has little overlap with the bulk states.

Figure R2 | Profile of the sum of probabilities of all bulk states. We use parameters of third sample where $\gamma = 0.705\pi$ and $\theta = 0.283\pi$.

2st Reviewer -- Comment 5:

5. Following Q4, the eigenstates in Figs. 4b and 4c have almost no overlap with each other, and thus one can selectively excite either state. I guess this (no overlap) is not a universal behavior. With some other

parameters, the corner states may not be completely isolated from the edge states, then one cannot selectively excite each one.

Response from Authors:

Actually, the topological corner states in our system are sub-lattice polarized, and occupy the corner site at the time frame we choose. The topological edge states are different, and they have less overlap with the corner sites. So, we can excite the corner states by a single site source. Certainly, if the corner states have strong overlap with the edge states, then it would be hard to selectively excite each one by single site source. In this case, we would need multiple sources to simultaneously excite different sites to efficiently excite different modes. Since the modes are orthogonal to each other, we can always selectively excite different modes by engineering the source profile.

GENERAL COMMENTS FROM 3st REVIEWER:

The paper "Time-periodic corner states from Floquet higher-order topology" by Zhu et al. provides an experimental demonstration of a Floquet HOTI exhibiting topological corner states protected by space-time symmetries. The experiment uses a 3D acoustic lattice with the z-axis playing the role of time. The authors claim that this is the first experimental realization of this theoretically proposed phenomenon, which makes the study valuable.

The main achievement of the paper is the experimental realization and observation of a HOTI. I, therefore, expect the advances reported on in the paper to be of interest mainly to experimentalists, but also to motivate theorists working on band topology. The experiment does not report anything exotic, or inconsistent with the underlying theory -- it merely provides an independent justification for HOTI in a specific physical system (3D acoustic lattice).

From a theoretical perspective, the paper does not provide a major new finding/discovery. However, a pioneering experimental realization itself can be valuable (and would justify publication in Nat Communications), provided it opens up a new avenue for further advances in the field.

Response from Authors:

We thank the reviewer for recognizing our work as a pioneering experimental realization.

SPECIFIC COMMENTS FROM 3st REVIEWER:

3st Reviewer -- Comment 1:

- The authors speak of a "hybrid topological protection induced by Floquet driving" -- I am a bit worried that the Floquet drive here is just a mathematical interpretation for the physics in the system. In reality, there is no time-periodic modulation applied to the system. Therefore, I urge the authors to do their best and avoid misleading formulations like the above, or at least make it clear that this is an interpretation.

Response from Authors:

We thank Reviewer #3 for the suggestion. We have revised this sentence as "...which constitutes a novel situation of hybrid topological protection" To further highlight this effective interpretation, we have stressed this point at the beginning of this paragraph as "Here, we experimentally demonstrate an acoustic Floquet HOTI exhibiting topological corner states protected by space-time symmetries (Fig. 1b). The time-

dependent drive is simulated by periodic modulation along a spatial axis in a static 3D lattice, which is a common method of realizing Floquet dynamics [14, 30].”

3st Reviewer -- Comment 2:

- Several times in the manuscript, the non-equilibrium character of the phenomenon is brought up. I would like to bring the authors' attention to the fact that not every time dependence results in non-equilibrium phenomena -- sometimes time is just an illusion of the reference frame. This is particularly easy to see in the present system where energy is conserved at all times [the system is a static 3D lattice with the z-axis playing the role of time]. I think all these statements should be corrected.

Response from Authors:

Thanks for the suggestion. We have removed the word of “non-equilibrium” in the manuscript.

3st Reviewer -- Comment 3:

- Although I am not an expert in the field of experimental acoustics, I am surprised by the way the experimental data is presented in the main text. E.g., in Fig 3 panels b,c are theoretical data, while panel d is supposed to be the experimental data. There are basically no units for the measurement, the color bar goes from zero to Max, and the Max value is not easy to find. This applies also to Fig 4. Naively I would expect experimental data to have noise/uncertainty, the effects of which can/should be discussed, etc.

Response from Authors:

We thank the reviewer for raising this concern. The unit of experimental data in our acoustic experiments is Pa. However, in most cases, such as plots of field distributions and frequency spectra, the data are usually normalized to the maximum value and the unit is omitted since only relative strength at different positions/frequencies matters. This is also the case for our plots for the experimental data. Similar handling of experimental data can be found in the literature (c.f., Fig. 4 in [*Nat. Mater.* 18, 113 (2019)], Fig. 3 in [*Nature* 560, 61 (2018)] and Fig. 1 in [*Phys. Rev. X* 6, 021007 (2016)]).

Regarding the experimental noise/uncertainty, we agree that they exist in any experiment. However, for acoustic measurement, in most cases the effect of noise/uncertainty is of no interest, for two main reasons. Firstly, current acoustic technologies allow very accurate and stable measurement (e.g., using Hi-Fi microphones, a routine device in acoustic experiments). Therefore, the noise/uncertainty from the equipment can be excluded. Secondly, in our experiment, the samples are stably positioned to minimize environmental vibrations and perturbations. The low level of noise/uncertainty in our results is also consistent with other papers in the literature dealing with similar experimental setups.

3st Reviewer -- Comment 4:

- Similar to the issue above -- the data for the spectrum presented in the main text (Fig 3a and Dif 4a) comes from a numerical simulation. The experimental results would make much more compelling evidence for the realization of a HOTI, if the spectrum were measured in a realistic experiment. I acknowledge the answer to the corresponding question by Reviewer 2, yet I urge the authors to seriously consider developing techniques or measuring the bulk spectrum in acoustic systems in future studies.

Response from Authors:

We thank the reviewer for the suggestion and fully agree measuring the bulk spectrum would be quite useful, not only to the present work but also to many future works. We will carefully consider developing new techniques in future studies.

3st Reviewer -- Comment 5:

- The study on the effect of disorder in the SI, Sec VI, can ideally be done in the experiment itself, through imperfections in the waveguide. So far Fig S8 is a theoretical simulation, and HOTIs have been analyzed by theorists before. It would be nice to test the robustness to disorder experimentally.

Response from Authors:

We thank the reviewer for the suggestion. However, it is technically difficult for us to precisely demonstrate the effect of disorder as in Fig. S8 in SI.

As Reviewer #1 pointed out, Fig. S8 requires a particle-hole symmetry, which, strictly speaking, does not exist in any acoustic system. Instead, our system satisfies a generalized particle-hole symmetry, $(k, z)C = k_z^0 - H^*(-k, z)$, which pins the momentum along z to be at k_z^0 or $k_z^0 \pm \pi/L$. The shift k_z^0 comes from the on-site potential of each acoustic waveguide, and is a constant at a fixed frequency. Therefore, our system is still robust.

Nevertheless, as we have clarified in the last response letter, it is extremely difficult to measure the Floquet (i.e., kz) spectrum. Probing a Floquet bandstructure at a specific kz is known to be a difficult problem in the field.

Actually, there is already some disorder in our samples. For example, the couplings in our system are realized with thin coupling channels with width 5 mm and separation 2 mm. However, the 3D stereolithography fabrication can only maintain a resolution around 0.1 mm. In other words, there is a roughly 2% uncertainty in width, or 5% uncertainty in separation, for the coupling channels. Therefore, our experiments have already demonstrated to some degree the robustness of predicted phenomena against moderate disorder.

Reviewers' Comments:

Reviewer #1:

Remarks to the Author:

I think my concerns have been well addressed and I'm glad to support the publishment of this manuscript.

Reviewer #2:

Remarks to the Author:

The authors have successfully addressed all my concerns in the last report, hence I will recommend it for publication.

Reviewer #3:

Remarks to the Author:

The authors have responded to all points I raised and implemented changes in the manuscript accordingly. I am happy with the changes and I believe the manuscript can now be published.